# Decoding Codon Bias: The Role of tRNA Modifications in Tissue-Specific Translation

**DOI:** 10.3390/ijms26020706

**Published:** 2025-01-15

**Authors:** Daisuke Ando, Sherif Rashad, Thomas J. Begley, Hidenori Endo, Masashi Aoki, Peter C. Dedon, Kuniyasu Niizuma

**Affiliations:** 1Department of Neurology, Tohoku University Graduate School of Medicine, Sendai 980-8575, Japan; daisuke.ando.a5@tohoku.ac.jp (D.A.); aokim@med.tohoku.ac.jp (M.A.); 2Department of Neurosurgical Engineering and Translational Neuroscience, Graduate School of Medicine, Tohoku University, Sendai 980-8575, Japan; kniizuma@tohoku.ac.jp; 3Department of Neurosurgical Engineering and Translational Neuroscience, Graduate School of Biomedical Engineering, Tohoku University, Sendai 980-8575, Japan; 4Department of Biological Sciences, University at Albany, Albany, NY 12222, USA; tbegley@albany.edu; 5Department of Neurosurgery, Tohoku University Graduate School of Medicine, Sendai 980-8575, Japan; hideendo@gmail.com; 6Department of Biological Engineering, Massachusetts Institute of Technology, Cambridge, MA 02139, USA; pcdedon@mit.edu

**Keywords:** tRNA modifications, queuosine, codon optimality, ribosome profiling, codon optimization, mRNA translation

## Abstract

The tRNA epitranscriptome has been recognized as an important player in mRNA translation regulation. Our knowledge of the role of the tRNA epitranscriptome in fine-tuning translation via codon decoding at tissue or cell levels remains incomplete. We analyzed tRNA expression and modifications as well as codon optimality across seven mouse tissues. Our analysis revealed distinct enrichment patterns of tRNA modifications in different tissues. Queuosine (Q) tRNA modification was most enriched in the brain compared to other tissues, while mitochondrial tRNA modifications and tRNA expression were highest in the heart. Using this observation, we synthesized, and delivered in vivo, codon-mutated EGFP for Q-codons, where the C-ending Q-codons were replaced with U-ending codons. The protein levels of mutant EGFP were downregulated in liver, which is poor in Q, while in brain EGFP, levels did not change. These data show that understanding tRNA modification enrichments across tissues is not only essential for understanding codon decoding and bias but can also be utilized for optimizing gene and mRNA therapeutics to be more tissue-, cell-, or condition-specific.

## 1. Introduction

The translation of the gene code to functional protein is at the heart of all processes of life. This multistep process, which starts with transcription and goes all the way to post-translational protein modifications, is under strict regulation at every step. At the heart of this process lies mRNA translation, which is one of the most complex and heavily regulated processes. mRNA modifications and structure [1,2], ribosome assembly and modifications [3], and tRNA epitranscriptome [4] are few of the factors that interplay to regulate proper mRNA translation and proteostasis. In recent years, there has been increased interest in the complexity and regulatory roles of the tRNA epitranscriptome on mRNA translation [4,5]. Importantly, alteration of tRNA expression [6,7], tRNA-derived small non-coding RNAs (tsRNAs) [8], and tRNA modifications [9] were shown to impact mRNA translation and play roles in diseases such as cancer, metabolic disorders, and neuropsychiatric disorders [7,8,9,10,11,12]. However, our understanding of how tRNA epitranscriptome regulates mRNA translation, while substantially improving over the past years, is far from complete.

A limited set of tRNA anticodons, 48 in humans and 47 in mice, decode 64 codons in the mRNA [13]. To compensate for this mismatch, non-cognate codon recognition at the wobble anticodon position, position 34 in the tRNA, allows for the decoding of the codons via the limited available tRNA pool [4,14]. This process is fine-tuned by tRNA modifications at the wobble position [4,5,15,16]. For example, tRNA modifications can expand or restrict the codon decoding by anticodons [4,16]. Some examples include adenosine-to-inosine conversion (A-to-I), which expands the recognition of NNU codons by ANN anticodons to NNU, NNC, and NNA by INN anticodons [4,17,18] and queuosine modifications that expand the recognition of NAC codons to NAY (NAC and NAU) codons [16,19,20,21].

Previous works have elucidated the dynamic nature of the tRNA epitranscriptome in regulating processes such as the stress response [6,15,22,23,24,25] and growth conditions [26] and how these dynamic changes impact codon recognition [6,25,27]. Nonetheless, much remains to be elucidated on the role of tRNA epitranscriptome in regulating physiological mRNA translation in various tissues and cells and their role in tissue physiology. Several groups have analyzed tRNA modifications [28] and tRNA expression [29,30] in mice tissues in order to elucidate such regulation. While these efforts did reveal important phenomena on tissue-specific expression of the tRNA epitranscriptome and its potential links to mRNA translation, they were limited in focusing on only one aspect of tRNA epitranscriptome at a time or not providing direct correlation with mRNA translation and codon decoding. In addition, the codon metrics used for the analysis in some works still require revisiting.

Here, we analyzed tRNA epitranscriptome (tRNA modifications and mature tRNA expression) from seven mouse tissues using sequencing and mass spectrometry [Figure 1a]. We combined our analysis with Ribo-seq analysis and applied various codon metrics to our dataset to reveal that tRNA modifications are linked to codon usage and optimality in various tissues. We further utilized the expression of tRNA modifications as an input for codon optimization of gene constructs to show how such information can be used to fine-tune mRNA delivery and viral-based gene therapeutics in a tissue- and cell-specific manner.

## 2. Results

➢
**tRNA and non-coding RNA modifications show tissue-level enrichment patterns**


We first began by exploring the tRNA modification landscape in each of the seven tissues used in this study [Figure 1a]. Using high-throughput liquid chromatography–tandem mass spectrometry (LC-MS/MS) analysis of tRNA-enriched small RNA fraction (>90% tRNA), we were able to detect 46 known mammalian modifications that occur in the tRNA or in other small non-coding RNAs (such as m6A and m6,6A) [Appendix A, Table 1] [4,31]. Our results revealed significant tissue specific enrichment (i.e., relative expression levels) of certain modifications as well as tissue and modification clustering patterns [Figure 1b, Appendix A]. All tested modifications were significant on one-way ANOVA statistical test (*p <* 0.05) with Tukey’s post-hoc analysis. Detailed graphs with multiple group comparisons are presented in Appendix A. A cluster of related modifications, mcm5U, mcm5Um, and ncm5Um, which are regulated by the same enzyme system [4,32], were most enriched in the brain as well as queuosine (Q) and its derivative mannosyl queuosine (manQ) but not its galactosyl derivative (galQ) [Appendix A]. Two tRNA modifications that occur in mitochondrial tRNA, and not cytosolic tRNA, were enriched mostly in the heart and muscles, namely, f5C and ms2i6A [33,34] [Appendix A]. f5Cm, which is downstream of f5C and occurs in cytosolic tRNA-Leu^CAA^ [33], was also enriched in the brain at comparable levels to the heart and muscles, while hm5C, which is upstream of f5C, was most enriched in the lung and muscles [Appendix A]. The liver and spleen had lower levels of almost all tRNA modifications; however, the spleen showed an enrichment in a specific set of modifications that include 2′O-ribose-modified nucleotides such as Cm, Gm, and Am, as well as several modifications not known to occur in the tRNA such as m6,6A and m6Am [4,31] but that do occur in other small non-coding RNAs [Appendix A].

Next, we directed our attention to how tissues cluster and correlate with each other, which can give an idea about the fine-tuning of tRNA modifications towards tissue specific functions. To that end, we conducted Pearson’s correlation analysis [Figure 1c] and partial least square regression discriminant analysis (PLS-DA) [Figure 1d] to understand such relations between tissues. Pearson’s correlation revealed that the brain had higher correlation with the kidney while it had a negative correlation with the spleen and lungs and no correlation with the liver [Figure 1c]. PLS-DA analysis [Figure 1d] confirmed the relationship between tissues, revealing that the brain clusters closely with the kidney while the lung, spleen, muscles, and heart form another cluster. Further analysis of the distribution of modifications in the PLS-DA plot [Appendix A], revealed how clusters of modifications correlated with tissue clusters. In addition, VIP analysis [Appendix A] revealed important contributors to each tissue cluster expression pattern. For example, m66A contributed to the spleen and lung clustering patterns while Q contributed to the brain, followed by kidneys. In addition to tissue clustering, we analyzed the correlation between tRNA modifications [Figure 1e]. Our analysis revealed clusters of modifications that follow the same tissue expression patterns. For example, methylation modifications, such as m1A, m5C, m3C, etc., generally clustered together, while 2′O-ribose modifications clustered together, and so on.

In summary, our analysis revealed significant tissue specific relative enrichment patterns of tRNA modifications. The variations observed in multiple modification systems that are regulated by the same enzymatic cascade allude to strict tissue-specific regulation that cannot be explained by variations in tRNA expression changes, given that these systems are confined to the same isoacceptors (such as the case of the hm5C/f5C system, Q/manQ/galQ system, and so on).

➢
**tRNA isoacceptors are stably expressed equally across tissues.**


tRNA modification levels could be altered not only due to changes in enzyme expression, but due to changes in tRNA expression or stoichiometric representation of the modifications. To resolve the question as to whether the changes are due to changes in tRNA expression, we analyzed the expression of mature tRNAs and their respective anticodons across tissues. Sequencing analysis revealed tissue specific variations in the content of tRNAs and various non-coding RNAs. For example, spleen and lung showed very low levels of tRNAs compared to other tissues, while they had significantly higher snRNAs and snoRNAs [Figure 2a]. This pattern was also evident when RNA samples were analyzed using Bioanalyzer during the regular quality control check performed prior to LC-MS/MS or sequencing. Spleen and lung consistently showed higher fractions of small RNA peaks between approximately 20 and 40 nucleotides in length [Appendix A]. Such patterns cannot be attributed to simple RNA degradation during extraction as they were operator-independent (replicated by two different operators) and were found in all samples analyzed. This finding can also explain the enrichment in non-tRNA modifications in the spleen and lung compared to other tissues. However, such lower levels of tRNA alone cannot explain the pattern of tRNA modifications observed in the lungs and spleen. For example, the lungs had comparable mcm5U levels to the brain [Appendix A]. The lungs also had the highest levels of hm5C [Appendix A] and ac4C [Appendix A]. The spleen had comparable levels of mcm5U and mcm5Um to the heart [Appendix A]. In addition, mitochondrial tRNAs (Mt_tRNA) were expressed in the heart at much higher levels, consistent with the enrichment of mitochondria related tRNA modifications in the heart [Figure 2a]. However, the muscle, despite showing higher levels of mitochondrial tRNA modifications, had comparable Mt_tRNAs to the brain and kidneys [Figure 2a]. We also observed notable differences in the ratios of tRNAs pertaining to specific amino acids in different tissues [Figure 2b]. Notably, selenocysteine (SeC) representation was higher in spleen and lung, which might be related to the known immune functions of SeC [36]. Next, we collapsed the mature tRNA counts by anticodon and calculated the expression of different tRNAs. As reported previously [29], the relative normalized expression of different tRNA isoacceptors within each tissue were relatively stable, which is reflective of the stability of tRNA housekeeping genes [29,37] [Figure 2c]. Thus, changes in tRNA isoacceptor relative ratios in different tissues cannot be the reason behind the differences observed in tRNA modification levels. Consequently, stoichiometric levels of tRNA modifications (i.e., the ratio of modified tRNA transcripts of an isoacceptors to the non-modified ones) are the most likely explanation for the differences between tissues. An example of this is Q, where it was shown previously using an orthogonal method to our LC-MS/MS approach that the stoichiometry of Q is highest in the brain and lowest in the spleen [19].

Mitochondrial tRNA isoacceptors displayed the same uniformity of distribution as their cytosolic counterparts [Figure 3a] despite the heart being several folds more enriched than other tissues [Figure 3b]. While this could explain the enrichment in mitochondrial tRNA modifications in the heart, it fails to explain the significant enrichment of f5C in the muscles, again signifying the role of stoichiometric levels of tRNA modifications in this context.

In summary, our analysis, in agreement with previous work [29], revealed almost uniform normalized expression of tRNA isoacceptors in different tissues, which is also in agreement with the notion of the existence of tRNA housekeeping genes that are stably expressed in all cells and tissues [37]. The sequencing data also reveal that the enrichment patterns of tRNA modifications across tissues cannot be solely explained on the grounds of tRNA expression. Rather, factors such as stoichiometric expression and enzymatic activity appear to play important roles.

➢
**Unique translational patterns observed in various tissues.**


To understand the potential links between the tRNA epitranscriptome and tissue-specific translational patterns and codon decoding, we conducted ribosome profiling (Ribo-seq). We were mainly interested in observing the clustering patterns at the translational level as well as conducting codon analytics to corroborate these data with our observations at the tRNA level. Heatmap clustering [Figure 4a, Appendix A] and tSNE [Figure 4b] revealed unique clustering of the brain vs. other tissues.

It was also evident that the clustering patterns observed at the translational level could faithfully replicate those observed at the tRNA modifications level [Figure 1c,d], where the brain was somewhat unique but closer to kidneys and liver, while the spleen and lung clustered closely together, as well as the heart and muscles. We next analyzed the differentially translated genes (DTGs) in each tissue versus all the others [Figure 4c] and observed various gene translational patterns. The brain showed the largest number of significant DTGs versus other tissues followed by the liver and heart. Indeed, this was also evident when we conducted detailed DTG analysis for tissue pairs [Appendix A]. Using the DTGs from each tissue versus all others, we generated a UMAP of Gene Ontology biological processes (GOBP) pathway analysis, which showed clustering for each tissue in terms of enriched pathways, with significant overlaps between closely related tissues (heart and muscle or lung and spleen) [Appendix A]. In addition, the activation matrix of GOBP pathways showed that unique pathway enrichment revealed the translational patterns of organ-specific pathways that could be attributed to organ function [Appendix A]. For example, pathways related to mitochondrial function and muscle contraction were enriched in the heart and muscles, in agreement with their known function, and corroborating the enrichment in tRNA modifications linked to mitochondrial activity, while pathways related to bile function and cholesterol and lipid homeostasis were enriched in the liver, etc. Finally, we evaluated whether the clustering and gene translation patterns in different tissues can be used to accurately annotate said tissues. The analysis revealed a good degree of success in annotating the tissues, further supporting the robustness of the sequencing dataset [Appendix A]. We further explored our Ribo-seq data for the expression of known tRNA- and small RNA-modifying enzymes, with a focus on tRNA-Q- and mcm5U-related enzymes [4] [Appendix A]. While we observed differences in expression across tissues, clustering, enzyme expression, as well as PLS-DA analysis did not correlate with what we observed at the level of tRNA modifications themselves [Appendix A]. Such findings preclude that the differences in tRNA modifications observed across tissues are due to differences in enzyme expression.

➢
**Tissue codon usage and optimality are governed by tRNA modifications levels.**


tRNA modifications are known to contribute to mRNA translation by altering codon recognition, usage, and optimality [4,16,19,25,27]. To elucidate the potential for such regulatory fine-tuning on tissue specific mRNA translation, we analyzed three metrics for codon analysis: isoacceptor codon frequency (which measures the expression of synonymous codons compared to each other in the coding sequence of a given mRNA), total codon frequency (which compares the expression of a codon with all other codons in the mRNA sequence) [38,39], and A-site pausing (which measures ribosome dwelling at each codon) [40] [Appendix A].

First, we started by analyzing the codon isoacceptor frequencies in each tissue using the top and bottom 200 expressed genes by normalized read counts [Figure 5]. As explained in the methods section, the number of synonymous codons were counted for each mRNA and compared to the genome average to generate T-statistics. T-stat value of >2 or <−2 indicates statistical significance (*p <* 0.05). Positive T-stat indicates higher enrichment while negative T-stat values signify the opposite. Heatmap clustering revealed a distinctive A/T- vs. G/C-ending codon bias with few exceptions [Figure 6a, Appendix A]. Except for the brain and the heart, all tissues were biased towards A/T-ending codons, evident from the increased usage of these codons in upregulated genes and their underusage in downregulated genes, in line with previously published work [41]. On the other hand, the brain and heart showed the lowest degree of bias and showed a shift towards G/C-ending codon selection. Pearson’s correlation coefficient analysis was conducted on the isoacceptor frequencies of the top and bottom 200 genes separately [Figure 5b]. It was evident that the differences in tissues were mainly apparent in the downregulated genes, with the brain being furthest from all tissues except the heart, in agreement with the observed codon biases. Further, PLS-DA analysis revealed unique clustering patterns akin to what was observed at the level of tRNA modifications and translation, with few exceptions. For example, the spleen and lung clustered together and were far from the brain [Figure 5c, Appendix A]. Based on the impression from our data that the brain is unique compared to other tissues, we focused our attention on two tRNA modification clusters that were highly expressed in the brain: tRNA-Q and its derivatives and the mcm5U cluster, its related modifications, and their corresponding codons. First, we explored the usage of Q-decoded codons and their correlation with the expression of tRNA-Q [Figure 5d]. tRNA-Q expands the codon decoding from NAC to NAY codons [15], which includes Asn-AAC/U, Asp-GAC/U, His-CAC/U, and Tyr-TAC/U. manQ decodes Asp-GAC/U while galQ decodes Tyr-TAC/U. However, Q was shown to enhance the decoding speed of NAC codons more than NAU codons [42], while it protects against ribosome frameshifting across NAU codons [43]. We observed a positive correlation between C-ending Q-codons and Q modifications, except for Tyr-TAC [Figure 5d]. This can be explained by the fact that Q and manQ were enriched in the brain but galQ did not show such enrichment [Figure 1b]. In addition, Tyr-TAC/U did not show strong bias in our analysis as compared to other Q-codons [Figure 4a]. It is also worth mentioning that the C-ending Q-codons, NAC codons, were negatively correlated with their synonymous NAU codons, signifying strong codon bias and selection [Figure 5e]. On the other hand, for mcm5U and mcm5s2U modifications, which are related to NAA codon recognition, mcm5U showed a positive correlation trend with GlnCAA and LysAAA, while mcm5s2U showed a negative correlation trend with LysAAA [Figure 5f], and the correlation coefficient for the other NAA codons was generally low, compared to Q-codon.

Next, we repeated our analysis at the level of total codon usage, which can give us clues as to the mRNA amino acids and sequence preferences [Figure 6]. In this analysis, we observed strong bias in all tissues between codon frequencies [Figure 6a], but with weaker A/T- vs. G/C-ending codon bias. Differences between tissues were apparent in the bottom 200 genes and not the top 200 genes when we conducted Pearson’s correlation coefficient analysis [Figure 6b]. In addition, PLS-DA analysis, while the tissues showed different clustering patterns compared to isoacceptors frequencies, still showed the brain to be unique while the lung and spleen clustered together [Figure 6c and Appendix A]. The correlation between Q-codons and mcm5U codons with their respective modifications was weaker or disappeared altogether, and there was no bias observed between isoacceptors in their total codon frequency counts [Figure 6d–f].

Finally, we examined the ribosomal A-site pausing, which gives an indicator of the speed of ribosome decoding across a given codon [Figure 7]. We compared the brain to each of the other tissues to elucidate the relative A-site pausing. Q-codons, except for Asp-GAC, were readily decoded in the brain compared to most other tissues, indicating increased optimality and decoding speed [Figure 7a,b]. mcm5U codons showed variable optimality, with NAA mcm5U codons being readily decoded, while Gly-GGA and Ala-GCA had variable optimality or were suboptimal altogether [Figure 7a,c]. The results from A-site pausing were mostly in agreement with isoacceptor codon frequency analysis, indicating that synonymous codon usage and bias are determinants of decoding efficiency and mRNA translation across tissues.

➢
**tRNA modifications inform codon optimization algorithms for gene therapy.**


The role of tRNA modifications in driving mRNA translation via dictating codon optimality have been extensively studied in the literature [4,11,15,16,25,27,44]. However, despite the wealth of information available on many modifications, the function of most tRNA modifications in a tissue/cell-specific context remains unknown. Indeed, validating all tRNA modifications’ influence on codon decoding in different tissues and in vivo is untenable in a single study. Nonetheless, the importance of understanding the role of tRNA modifications at the tissue level cannot be overstated. Thus, we asked whether the knowledge of tRNA modification levels can inform codon optimization algorithms for optimal protein expression after gene or mRNA therapy. We focused on tRNA-Q modifications, given the clear enrichment of tRNA-Q in the brain vs. other tissues [Figure 8a]. tRNA-Q expands the codon decoding from NAC codons to NAC and NAU codons (NAY codons) [20,21,42] [Figure 8b]. Based on the data from our analysis, we focused on the brain (high Q levels) and liver (low Q levels) as example tissues to test the hypothesis. We argued that NAC codons would be decoded efficiently in either tissue, due to the existence of cognate anticodons. However, NAU codons will not be efficiently decoded in the liver due to lower Q levels. To test our hypothesis, we generated adenoviral vectors (AAV vectors) carrying wild-type (WT) or mutant EGFP [Figure 8c]. EGFP codon sequences were mutated from the original optimal NAC codons to NAU codons for each of the Q-modified amino acids tRNAs, or all four together [Appendix A]. We used tdTomato as internal control. AAV vectors were injected intravenously into mice, and tissue samples were collected after 3 weeks for Western blotting validation of protein levels. In the brain, we did not observe changes in EGFP protein expression after mutating the codons [Figure 8d]. However, in the liver, we observed statistically significant downregulation of EGFP protein expression after mutating the Q-decoded codons from NAC to NAU. This observation provides an indicator that knowledge of tRNA modification levels could serve as an input for codon optimization strategies and can be further used for improving gene therapy efficiency. This also provides proof that the levels of tRNA modification contribute to tissue-specific codon decoding and to the observed codon optimality across tissues.

## 3. Discussion

Over the years, the role of tRNA in fine tuning codon usage and optimality has been increasingly appreciated [4,5,11]. The ability of tRNA, either via tRNA expression or tRNA modifications, to shift translation towards sets of genes via altering codon optimality has been shown to play important roles in conditions such as cancer and oxidative stress [6,15,25,44]. In particular, the role of tRNA modifications at the wobble position in expanding or restricting codon decoding has been shown to be vital in regulating mRNA translation [4]. The importance of tRNA modifications becomes apparent when one considers that a limited set of anticodons (48 in humans) need to decode for 64 mRNA codons. Nonetheless, the physiological role of tRNA modifications in regulating adaptive translation to meet the needs for different cells remains unclear. Importantly, most research on tRNA modifications has either focused on cancer biology or used cell lines, which do not reflect primary cells’ physiology. In this work we present a comprehensive analysis of the tRNA epitranscriptome and identify its role in regulating codon usage and optimality across tissues. While several previous works have attempted to tackle this subject from various angles [28,29,30], to the best of our knowledge, our work represents the first comprehensive analysis of the various layers of the tRNA epitranscriptome and their direct correlation to translatomes. We also validated the links between tRNA modifications and codon decoding in vivo using mutated EGFP constructs, which, to the best of our knowledge, was not performed previously in mammals. This work also provides proof of concept (PoC) data showing that knowledge of tRNA modifications in a tissue-, cell-, or cell state-specific context could be used to optimize gene therapies by fine-tuning the protein production in vivo.

At the tRNA modification levels, previous works by Guo et al. [28] showed tissue-specific enrichment levels but did not include the analysis of many anticodon modifications that are of great importance in fine-tuning codon optimality in tissues such as mcm5U, queuosine, or mitochondrial tRNA modifications. Further, previous works did not directly correlate tRNA epitranscriptome with codon decoding and translation [28,29] or only used limited metrics for codon analysis [30]. Here, we directly analyzed the correlation between the tRNA epitranscriptome on mRNA translation and codon decoding using multiple metrics. Our work shows, in line with the wealth of literature on tRNA modifications, that the expression of various modifications in different tissues could be linked to specific translational patterns and physiological functions of these tissues. Our data provide compelling evidence that tissue-specific codon optimality and decoding capacity are linked to tRNA modifications levels.

tRNA modification levels are governed by multiple factors. tRNA expression, stoichiometric levels of modification, and enzyme activity all contribute to dictating the level of tRNA modifications in each tRNA transcript. In our data, it was clear that neither enzyme levels nor tRNA isoacceptors’ relative ratios are major determinants of tissue modifications levels. Our data highlight the near uniformity of tRNA isoacceptor (anticodon) expression across tissues, in agreement with previous works [29]. While it was shown previously by Pinkard et al. [29] that there are tissue-specific isodecoder tRNA level differences, tRNA isoacceptor expression was very stable across tissues. This agreement between our work and Pinkard et al.’s [29] work, despite using different tRNA sequencing techniques, validates their findings, as well as ours. Importantly, while tRNA isoacceptors were shown to dynamically change in pathologies such as oxidative stress [6] and cancer [7] to impact codon decoding, it appears that this role is more or less stable in physiology. This was further shown in recent work by Gao et al. [37], where they showed that tRNA transcripts (isodecoders) vary between cell lines, but the anticodon pools (isoacceptors) are stable.

Codon usage and optimality have been a subject of immense interest, given their role in regulating many molecular processes such as mRNA stability and decay [45,46], ribosome decoding fidelity [47], and mRNA translation efficiency [5,48]. Previous works revealed a plasticity of codon optimality that adapts to cell states. For example, during oxidative stress, tRNA modifications and tRNA expression dynamically change to alter codon optimality and drive translation of antioxidant proteins [6,15,25]. This also occurs during cell proliferation and differentiation [49,50]. These findings alluded to the potential of tissue-specific codon optimality to regulate mRNA translation and local proteostasis. Such an idea was challenged previously in Drosophila tissues by Allen et al. [51]. In their work, Allen et al. [51] used codon-mutated probes to evaluate the expression of rare codons in various tissues. They found that the testis and brain can express rare codons compared to other tissues. Attempts to elucidate tissue-specific codon optimality in mammalian tissues have been limited, and mostly relied on published datasets [52] or public databases [41]; thus, the analysis was usually limited by the data available. For example, no formal correlation between tRNA pools or tRNA modifications was made to explain codon optimality patterns or specific phenomena observed at the codon level in previous published works [41,52], especially that the reliance on tRNA genes does not reflect actual tRNA transcript expression [13]. In addition, the reliance on codon adaptation index (CAI) or tRNA adaptation index (tAI) might also not be ideal in such contexts where tissue specific metrics are needed, given that these two metrics depend on genomic information and not actual mRNA or tRNA levels [53,54]. To the best of our knowledge, the work presented herein represents the first analysis of mammalian codon usage and optimality that relies on tissue-specific translational levels to deduce codon optimality. Previously, Benisty et al. [41] showed that most abundant mRNAs are A/T-ending codon biased. We observed the same pattern of enrichment in most tissues except for the brain and heart. However, while they made the argument that this bias is driven by tRNA expression profiles, we disagree on this notion, mainly because we observed differences in codon decoding across tissues that cannot be explained by the stable expression of tRNA isoacceptors that we, and others [29,37], observed. In addition, Benisty et al. [41] retrieved their tRNA data from another publication [55], which utilized TCGA datasets to deduce tRNA expression. We cannot deem the methods used to generate small RNA-seq datasets that were reported in the TCGA database to be fully capable of capturing full-length tRNA transcripts, especially as the methods used to generate these small RNA-seq datasets were not optimized to map mature tRNAs. Thus, while we cannot negate the role of tRNA transcripts completely in driving the global A/T-ending codon bias, that role cannot explain the tissue-specific patterns we observed, which could be better explained by tRNA modifications.

Codon optimization for gene and mRNA therapeutics have become a hot topic of research given the recent advances in mRNA vaccines, especially during the COVID-19 era [56]. Codon optimization was largely used to promote the stability and translation of bacterial and viral transcripts used for vaccination in mammalian cells [56]. In that sense, genome-based metrics, such as the CAI (codon adaptation index) [54] or tAI (tRNA adaptation index) [53], could be appropriate. However, we used a very different approach. We hypothesized that knowledge of the synonymous codon usage, driven by tRNA modifications, could allow for the tuning of protein expression of delivered gene constructs in different tissues. Indeed, we show here that knowledge of Q levels in the liver and brain allowed for just that. This opens the door to optimizing gene and mRNA therapeutics in ways that allow for maximum protein production in target tissues and cells while reducing the side effects from aberrant protein expression in other non-target tissues. This notion is particularly interesting when we consider that diseased tissues have codon usage patterns different from healthy tissues, evident by a wealth of literature [11,25,44]. Thus, proper characterization and understanding of the relationship between tRNA epitranscriptome and codon usage in diseases is of utmost importance to develop highly specific gene therapies.

tRNA modifications have been recognized as important players in many diseases and conditions [9,11,57]. Nevertheless, their cell-specific physiological roles are not explored for most of the known modifications. In this work, we show a wealth of tissue-specific enrichment of various modifications. While we could attribute that enrichment to specific codon optimality patterns or tissue functions (such as enrichment in mitochondrial modifications in the heart and muscles), it is untenable to validate the role of 40+ modifications in a single work. Thus, we focused our attention on Q modification, which has been a subject of interest lately, with multiple groups reporting its role in oxidative stress, brain health, and codon decoding [15,16,19,20,21,25,42,58]. We identified queuosine (Q) to be preferentially enriched in the brain and drive NAY codon recognition. Cirzi et al. also showed that Q enrichment was highest in the brain and lowest in the spleen using an orthogonal approach to ours, APM gel electrophoresis, which can reveal the stoichiometric enrichment in Q [19]. Q was first identified in E. coli in a series of studies in the 1960s [59,60,61]. Since its discovery, its role in regulating codon decoding, protein synthesis, and mitochondrial function have been studied by various groups. Q was shown to extend the codon decoding of GUN codons from NAC to NAC + NAU [20,62,63]. Q was also shown to be essential for preventing ribosome +1 frameshifting across NAU codons [43]. Q modifications were shown to play important role in regulating cellular response to arsenite induced oxidative stress and mitochondrial stress and dysfunction [15,25], as well as playing a role in hippocampal neuron function and cognition [19]. Loss of Q via gene editing of the TGT complex leads to ribosome stalling at the NAU codons [21,25]. Interestingly, despite the reported importance of queuosine in codon decoding in vitro [20,25], knockout mice were phenotypically normal apart from cognitive and memory dysfunction [19]. However, it was shown by the Farkas group that queuine (the precursor for Q in mammalian cells)-depleted mice, which are also deficient in Q, display a normal phenotype unless tyrosine is also withdrawn from the diet [64,65]. This distinct phenotype of Q deprivation indicates a context or neuronal subtype-specific function, in line with previous works that showed that Queuine, the precursor for queuosine, is protective against neurodegenerative diseases [66]. This neural-specific phenotype also sheds light on tissue- and cell-specific physiological functions of tRNA modifications that are rarely addressed. In the case of queuosine, it appears that the enrichment in the brain is vital for proper neuronal functioning, evidenced by Cirzi et al.’s work [19]. Such observations highlight the importance of understanding cell- and tissue-specific dependencies on tRNA modifications for proper translation in understanding tissue development, physiology, and pathologies.

In conclusion, we show that tRNA modification expression levels in various tissues are fine-tuned to tissue-specific codon decoding and mRNA translation requirements to maintain the physiological proteostasis needed for tissue functioning. We also show that knowledge of tRNA modification levels and codon optimality in specific contexts is essential for better gene and mRNA therapeutic development. It is important to note that this work was somewhat limited, as we did not conduct transcript specific tRNA modification analysis; thus, it remains unclear whether a modification that occurs in multiple tRNAs is enriched in all these tRNAs in each tissue or if there is some degree of fine-tuning. We also did not assess the levels of tRNA aminoacylation, which are known to play important roles in dictating translation and codon decoding [67]. Future research should attempt to replicate our findings using orthogonal methodologies to ensure that these results are not due to specific technical limitations. It is also important to further clarify the physiological roles of tRNA modifications in representative primary cells and not in commercially available cell lines that might not reflect the true physiology of tissues/cells.

## 4. Methods

### 4.1. Animal and Sample Collection and RNA Isolation

Eight-week-old male C57BL/6J mice purchased from Kumagai Shigeyasu Shoten (Sendai, Japan) were maintained on a 12 h light–dark cycle. For LC-MS/MS analysis, four mice were perfused with ice-cold PBS, and tissues (brain, lung, heart, liver, spleen, kidney, and muscle) were extracted immediately after perfusion and flash-frozen in liquid nitrogen until RNA extraction. Tissues were homogenized with QIAzol (QIAGEN, Hilden, Germany, #79306) and small RNA was isolated using a Purelink miRNA Isolation Kit (Thermo Fisher, Waltham, MA, USA, #K157001). For small RNA sequencing and ribosome profiling, three mice were perfused using ice-cold phosphate-buffered saline with 100 µg/mL cycloheximide (Sigma, St. Lousi, MO, USA, #01810). Tissues were extracted immediately after perfusion and flash-frozen until RNA extraction. Tissues were homogenized with lysis buffer (20 mM Tris-Cl pH 7.5, 150 mM NaCl, 5 mM MgCl_2_, 1 mM dithiothreitol (Sigma, St. Lousi, MO, USA, #D9779), 100 µg/mL cycloheximide, and 1% Triton X-100) and centrifuged at 13,000× *g* at 4 °C for 15 min. For small RNA sequencing, 100 µL of supernatant was transferred to fresh tubes, and then 900 µL of QIAzol was added. Subsequently, small RNA was isolated using a Purelink miRNA Isolation Kit, according to the manufacturer’s protocol. For ribosome profiling, 500 µL of supernatant was kept for the downstream procedure. All animal experiments were conducted according to procedures approved by the animal care facility of Tohoku University and according to the ARRIVE (Animal Research: Reporting In Vivo Experiments) guidelines. Ethical board approval was acquired prior to the commencement of this project (Approval protocol ID: 2022-Medical-103-03. Approval date 10 March 2023). No animals were excluded from the analysis. No randomization or blinding was performed.

### 4.2. Small RNA Quality Control

NanoDrop One was used to analyze RNA concentration and purity. RNA integrity was analyzed using Bioanalyzer 2100 and the Agilent bioanalyzer small RNA kit (Agilent, #5067-1548). Example bioanalyzer traces are provided in Appendix A.

### 4.3. Quantitative Analysis of tRNA Modifications by Mass Spectrometry

Quantitative liquid chromatography–tandem mass spectrometry (LC-MS/MS) analysis of RNA modifications was conducted as reported previously, with modifications [68]. Small RNA-enriched RNA fractions, which contained around 90% mature tRNA, were digested for 6 h at 37 °C using a digestion mixture of MgCl_2_ 2.5 mM, Tris (pH 8) 5 mM, Coformycin 0.1 μg/mL (Sigma, St. Lousi, MO, USA, #SML0508), Deferoxamine 0.1 mM (Sigma, St. Lousi, MO, USA, #D9553), Butylated hydroxytoluene 0.1 mM (Sigma, St. Lousi, MO, USA, #W218405), Benzonase 0.25 U/μL (Sigma, #E1014-25KU), Calf intestinal alkaline phosphatase (CIAP; Sigma, St. Lousi, MO, USA, #P5521) 0.1 U/μL, and Phosphodiesterase I (PDE I; Sigma, St. Lousi, MO, USA, #P3243) 0.003 U/μL. Samples and standards were injected into a Waters BEH C18 column (50 × 2.1 mm, 1.7 µm; Waters, Stamford Avenue, Altrincham Road, Wilmslow, UK, #186002350) coupled to an Agilent 1290 HPLC system and an Agilent 6495 triple-quad mass spectrometer. The LC system was conducted at 25 °C and a flow rate of 0.3 mL/min. Buffer A was composed of 0.02% formic acid (FA) in DDW. Buffer B was composed of 0.02% FA in 70% Acetonitrile. The buffer gradient is shown in Appendix A. The UPLC column was coupled to an Agilent 6495 triple-quad mass spectrometer with an electrospray ionization source in positive mode with the following parameters: gas temperature—200 °C; gas flow—11 L/min; nebulizer—20 psi; sheath gas temperature—300 °C; sheath gas flow—12 L/min; capillary voltage—3000 V; nozzle voltage = 0. Dynamic MRM was used to detect modifications using transitions and collision energies listed in Appendix A. Peak areas were normalized to the sum of the UV signal of the canonical nucleotides (U, C, G, and A) and expressed as area ratio. Four biological replicates were used per group.

### 4.4. Small RNA and tRNA Sequencing

We used methods that were described previously [69,70], with some modifications. Briefly, to remove 3′ conjugated amino acids, 500 ng RNA sample was deacylated in 0.1 M pH 9.0 Tris-HCl buffer at 37 °C for 1 h. Deacylated RNA was treated with AlkB demethylase (rtStar tRNA-optimized First-Strand cDNA Synthesis Kit, ArrayStar, Rockville, MD, USA, #AS-FS-004) at ambient temperature for 2 h to reduce tRNA methylation. RNA was purified using a Zymo Directzol RNA micro kit (Zymo research, Irvine, CA, USA, #R2060) and end-repaired with T4 Polynucleotide Kinase (T4 PNK; NEB, Ipswich, MA, USA, #M0201) at 37 °C for 30 min. All treated RNA samples were used for library preparation using a NEBNext Multiplex Small RNA Library Prep Set for Illumina (NEB, Ipswich, MA, USA, #E7300/7580) according to the manufacturer’s protocol. Size selection of amplified cDNA library was performed using 6% TBE PAGE in a range of 140 to 210 bp (microRNA to mature tRNA). Library concentration and quality were assessed by Bioanalyzer 2100 using a DNA 1000 chip (Agilent, Santa Clara, CA, USA, #5067-1504). All the samples were pooled and sequenced by Illumina Hiseq-X ten instrument in a pair-end, 150 bp read. Three biological replicates were used per group.

### 4.5. Small RNA Sequencing Data Analysis

Small RNA libraries were first processed by tRAX software (https://trna.ucsc.edu/tRAX/) accessed on 7 January 2025 [71] to perform adaptor trimming and alignment to mouse tRNA reference (mm10 genome) as well as quality control. tRAX was used to analyze mature tRNA and anticodon expression.

### 4.6. Ribosome Profiling

Ribosome profiling was performed as we previously reported [72], which was based on another protocol with a few modifications [73]. Ribosome foot-printing was performed on samples collected as mentioned above by adding 1.25 U/μL RNase I (NEB, Ipswich, MA, USA, #M0243L) to 500 μL clarified lysate and incubating samples on a rotator mixer for 45 min at room temperature. TRIzol reagent was added, and RNA was extracted using a miRNeasy Mini Kit (QIAGEN, Hilden, Germany, #217004). Ribosome-protected fragments (RPFs) were selected by isolating RNA fragments of 27–35 nucleotides (nt) using TBE-Urea gel. rRNA depletion was conducted using NEBNext rRNA depletion kit v2 (NEB, Ipswich, MA, USA, #E7400L) followed by end-repair using T4 PNK after purification of the rRNA depleted samples using Oligo Clean & Concentrator Kits (Zymo research, Irvine, CA, USA, #D4060). The preparation of sequencing libraries for ribosome profiling was conducted via the NEBNext Multiplex Small RNA Library Prep kit for Illumina according to the manufacturer’s protocol (NEB, Ipswich, MA, USA, #E7300S). Pair-end sequencing reads of size 150 bp were produced for Ribo-seq on the Illumina Hiseq X-ten system. Three biological replicates were prepared per group.

### 4.7. Ribosome Profiling Data Analysis

Quality control for Raw Fastq was performed using FastQC. Next, adapter trimming and collapsing of pair-end reads into one was performed using Seqprep. Trimmomatic [74] was used to further clean low-quality reads. Bowtie2 [75] was used to align the reads to a reference of rRNA and tRNA genes (mm10) to remove contaminants. After that, reads were aligned to the genome (mm10 downloaded from UCSC), reads were counted using FeatureCounts [76], and differential expression was conducted using Limma [77]. Omics Playground [78] was used for Ribo-seq data visualization and clustering as well as read normalization. Of note, we also used Stringtie [79] to generate TPKM normalized read counts that were comparable to the log2CPM generated from Omics Playground; thus, we performed downstream analysis using the Omics Playground output.

### 4.8. Codon Analysis

Isoacceptor frequency and total codon frequency were analyzed as previously reported [38,39]. The amino acid-normalized codon frequency is calculated as$$\mathrm{frequency}\mathrm{of}\mathrm{codon}i\mathrm{in}\mathrm{gene}j=\frac{\mathrm{number}\mathrm{of}\mathrm{occurences}\mathrm{of}\mathrm{codon}i\mathrm{in}\mathrm{gene}j}{\sum_{i=1}^{n}{\mathrm{number}\mathrm{of}\mathrm{occurences}\mathrm{of}\mathrm{isoacceptor}}_{ij}}$$
where isoacceptor refers to the synonymous codons for the amino acid encoded by codon *i* and n refers to the number of isoacceptors for a given amino acid (2 ≤ n ≤ 6).

The total codon frequency is calculated as$$\mathrm{Total}\mathrm{frequency}\mathrm{of}\mathrm{codon}i\mathrm{in}\mathrm{gene}j=\frac{\mathrm{number}\mathrm{of}\mathrm{occurences}\mathrm{of}\mathrm{codon}i\mathrm{in}\mathrm{gene}j}{\sum_{i=1}^{n}\mathrm{number}\mathrm{of}\mathrm{occurences}\mathrm{of}\mathrm{codon}i\mathrm{in}\mathrm{gene}j}$$
where n refers to the total number of codons used in the gene.

The *t*-statistic describing the codon frequency was calculated as$$\mathrm{T-}\mathrm{statistic}\mathrm{for}\mathrm{codon}i=\frac{x_{i}-{µ}_{i}}{s_{i}/\sqrt{n}}$$
where x_i_ refers to the mean frequency for codon *i* in the sample, ${µ}_{i}$ refers to the mean frequency for codon *i* across the Mus musculus genome, s_i_ refers to the standard deviation of the frequency for codon *i* in the sample, and n refers to the sample size.

A-site pausing was calculated using Ribotoolkit [40], where the brain was compared to each tissue to deduce the relative ribosome pausing at each codon. Next, the output was converted into log2 fold change for data representation.

### 4.9. Generation of Adeno-Associated Viral (AAV) Vector Plasmids

The original EGFP (WT) and mutant EGFP with codons corresponding to each tRNA-Q modification (Tyr, His, Asn, Asp), or all four codons (All) in an EGFP sequence modified from NA**U** to NA**C** were synthesized (Eurofins Genomics, Tokyo, Japan). To construct the vectors, CMV promotor and tdTomato sequences were obtained from pMuLE ENTR CMV tdTomato L5-L2 [80] (Addgene, Watertown, MA, USA, #62152), and CMV promotor for EGFP was obtained from pMuLE ENTR CMV eGFP L3-L2 [80] (Addgene, Watertown, MA, USA, #62143) by PCR amplification. All fragments were inserted into AAV backbone vector (pAAV-CAG-GFP, Addgene, Watertown, MA, USA, #37825) using NEB HiFi DNA Assembly Master Mix (NEB, Ipswich, MA, USA, #E2621), following the manufacturer’s protocol. The sequences for EGFP and mutated EGFP are presented in Appendix A.

The following primers were used for CMV promoter and tdTomato cloning:CTAGGAAGAGTACCATTGACGACATTGATTATTGACTAGTTATTAATCAATGTCTAGAGGCTCGAGCTCTT

The following primers were used for CMV promoter cloning:CGAGCCTCTAGACATTGATTATTGACTAGTTATTGGTGGCTTTAGGATCCGA

The following primers were used for AAV backbone cloning:TCGATATCAAGCTTATCGATAATCAGTCAATGGTACTCTTCCTAG

### 4.10. Viral Vector Production

Recombinant AAV vectors were produced through triple-transfection of vector plasmids pUCmini-iCAP-PHP.eB [81] (Addgene #103005) and pAdDeltaF6 (Addgene #112867) into HEK293T cells using 0.1% PEI MAX, pH 7.0 (Polysciences, Inc., Warrington, PA, USA, #24765-100). Viral vectors were purified by ultracentrifuge as previously described [82]. Briefly, cells were cultured with serum-free medium, and supernatant was collected 5 days after transfection. After virus concentration by PEG 8000 (Sigma, #P5413) and Benzonease treatment (Millipore, Burlington, MA, USA, #70746-3CN), virus was purified by ultracentrifuge using Optiprep (Serumwerk Bernburg, Benrburg, Germany, #1893) discontinuous gradient. AAV titer was determined by qPCR method using AAVpro Titration Kit (for Real Time PCR) Ver.2 (Takara-bio, Kusatsu, Shiga, Japan, #6233) following the manufacturer’s protocol.

### 4.11. In Vivo Virus Transduction

Six-week-old C57BL/6J male mice were injected intravenously with 1.0 × 10^11^ vector genome of AAV vector. Mice were sacrificed 3 weeks after transduction and tissues were extracted and flash frozen.

### 4.12. Western Blot

Tissue samples were collected and lysed with T-PER Tissue Protein Extraction Reagent (Thermo Fisher, Waltham, MA, USA, #78510) containing cOmplete protease inhibitor cocktail (Roche, Basel, Switzerland, #4693116001). The tissue lysate was centrifuged at 16,000× *g* for 20 min and supernatant was extracted for protein concentration measurement by Pierce BCA protein assay (Thermo Fisher, Waltham, MA, USA, #23227). Each amount of protein was loaded into 8–16% Mini-PROTEIN TGX Precast Protein Gels (Bio-Rad, Hercules, CA, USA, #4561106) and then transferred to Trans-Blot Turbo Mini 0.2 µm PVDF Transfer Packs (Bio-Rad, Hercules, CA, USA, #1704156). The membranes were blocked by 5% skim milk (GFP; Wako, #198-10605) or PVDF Blocking Reagent for Can Get Signal (RFP; TOYOBO, #NYPBR01). Then, they were incubated overnight with primary antibodies GFP (Abcam, Cambridge, UK, #ab6556) and RFP (MBL, Tokyo, Japan, #PM005) at 4 °C. The band was detected with HRP-conjugated anti-rabbit antibody (CST, Danvers, MA, USA, #7074S) and Pierce ECL Western Blotting Substrate (Thermo Fisher, Waltham, MA, USA, #32106). The protein bands were visualized by ChemiDoc MP (BioRad Hercules, CA, USA). The detected bands were analyzed using Image Lab Software (V 6.1, BioRad Hercules, CA, USA). One-way ANOVA with Dunnett’s multiple analysis and visualization was performed by GraphPad Prism 10 (GraphPad software).

### 4.13. Data Visualization and Statistical Analysis

Heatmaps and clustering and Pearson’s correlation analysis were conducted using Morpheus (https://software.broadinstitute.org/morpheus) 7 January 2025. Hierarchical clustering was performed using the “one minus Pearson correlation” metric and “average” as the linkage method. PLS-DA was conducted using the *R* language package mixOmics 6.26.0 [83]. Standard score (Z-score) analysis was conducted in *R*. Z-score was defined as the number of standard deviations by which the observed value (either in log2CPM and TPM in sequencing datasets or normalized peak area in LC-MS/MS) is above or below the mean value of what is being measured. ANOVA with Tukey’s post hoc analysis was conducted using SPSS v20 (IBM corp.).

## Figures and Tables

**Figure 1 ijms-26-00706-f001:**
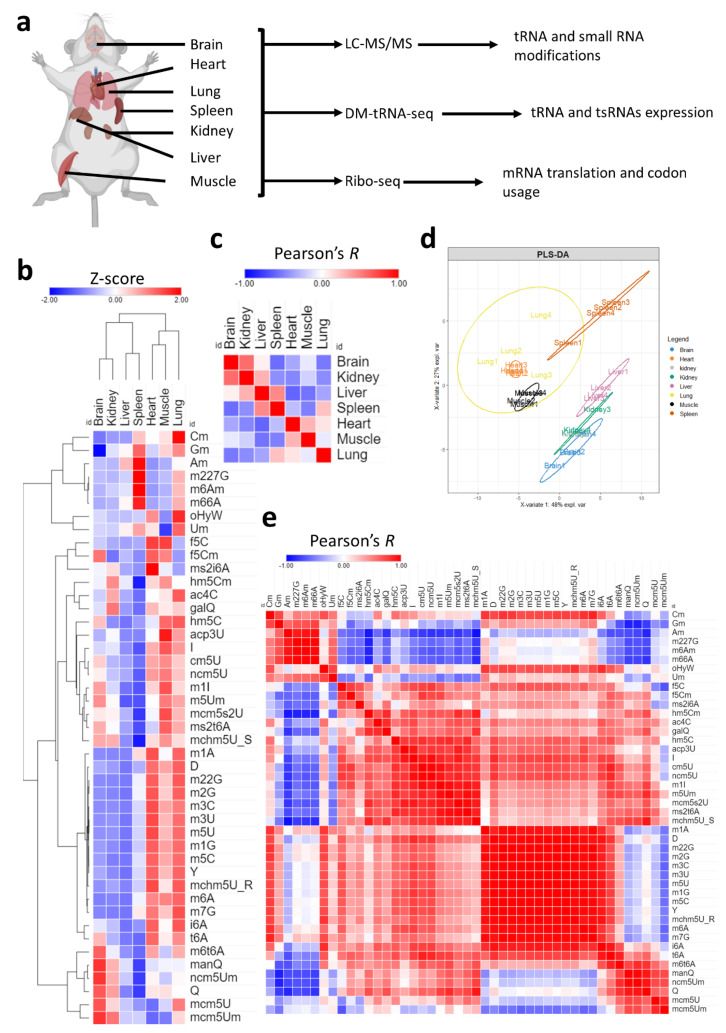
Analysis of tRNA modifications landscape across tissues. (**a**): Schematic showing the tissues analyzed and methods used in this study. Four biological replicates were used for LC-MS/MS analysis and three biological replicates were used for each sequencing method. (**b**): Expression of 45 different modifications across the 7 tissues. Data presented as Z-scores across tissues. N = 4 biological replicates per group. Detailed graphs with ANOVA and post-hoc analysis are presented in Appendix A. Normalized peak areas used for analysis are presented in Appendix A. (**c**): Pearson’s correlation between different tissues using Z-scores of tRNA modifications as input. The coloring represents values of Pearson’s coefficient (Pearson’s *R*) from −1 to 1. (**d**): PLS-DA analysis of tissue clustering patterns using normalized peak areas of tRNA modifications as input. (**e**): Pearson’s correlation across modifications.

**Figure 2 ijms-26-00706-f002:**
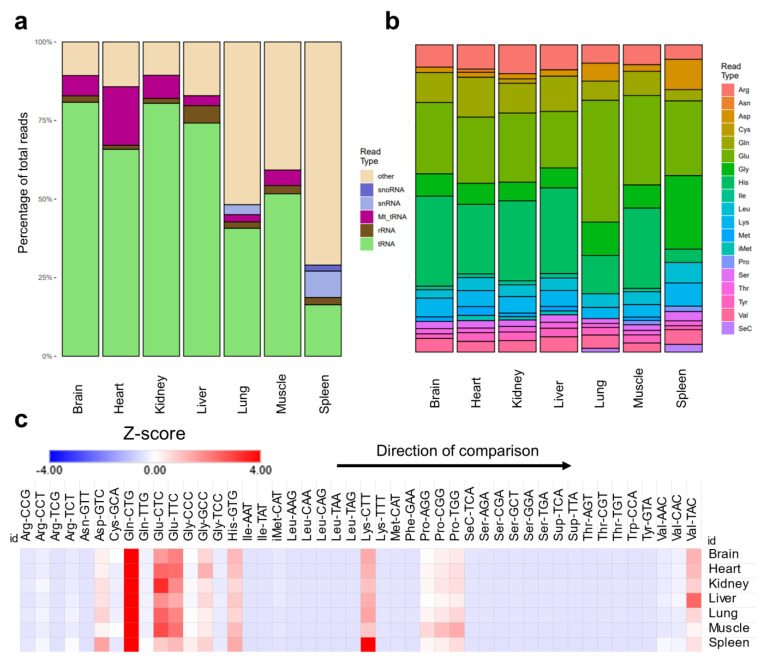
Analysis of mature tRNA expression across tissues. (**a**): Distribution of detected small non-coding RNAs in the sequencing datasets. (**b**): Distribution of different amino acids representing tRNAs in the datasets. (**c**): Relative ratios of tRNA anticodons (isoacceptors) in each tissue examined in our dataset normalized by their normalized read counts and represented as Z-score. Here, Z-scores are calculated within each tissue separately to show the distribution of isoacceptors within each tissue. This allows for the normalization across tissues with different tRNA read counts. Z-scores were used to represent normalized read counts to provide comparable heatmaps to Figure 2c. N = 3 biological replicates.

**Figure 3 ijms-26-00706-f003:**
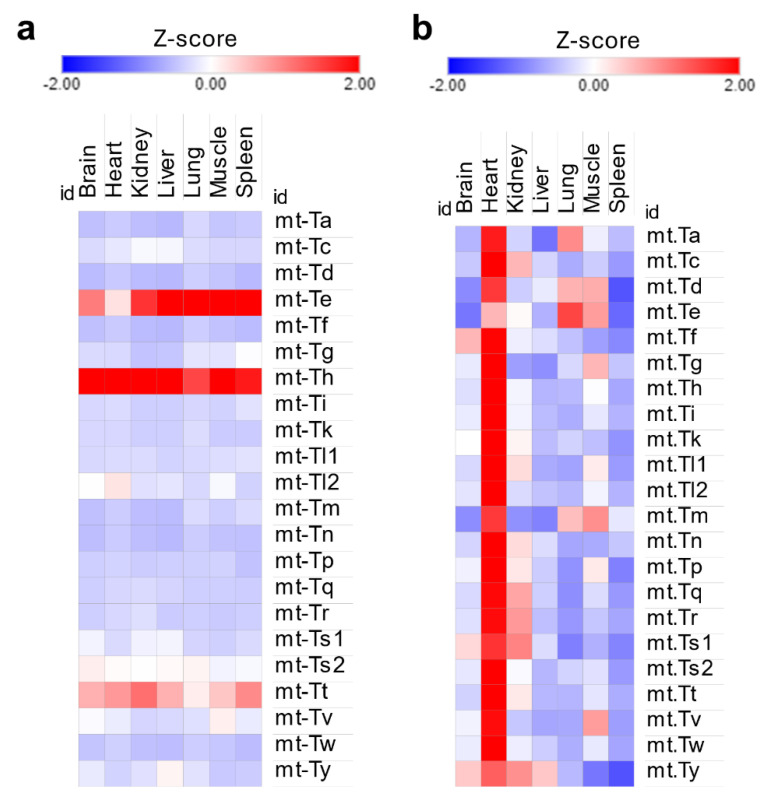
Analysis of mitochondrial tRNA expression in the tRNA-seq dataset. (**a**): Mt-tRNA isoacceptors are expressed stably in each tissue. Z-scores were calculated using normalized read counts for each tissue separately. (**b**): Z-scores of Mt-tRNAs expression across tissues. Here, Z-scores are used to represent differences in normalized read counts between tissues. The heart has the highest read counts of Mt-tRNAs.

**Figure 4 ijms-26-00706-f004:**
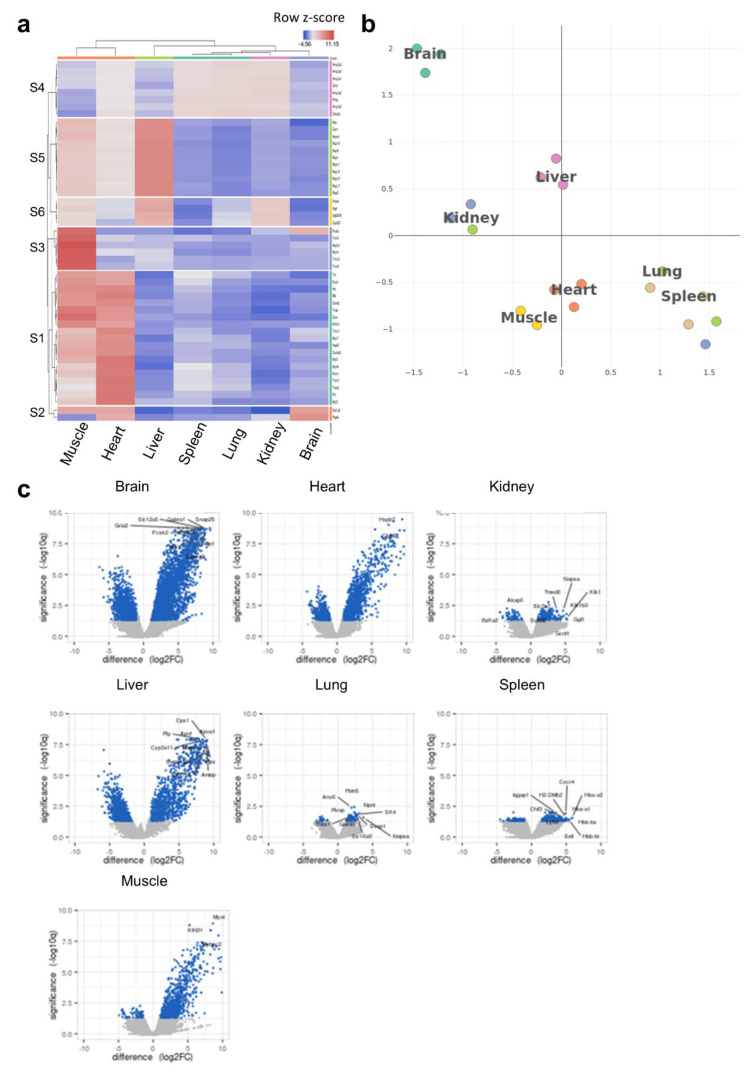
Ribo-seq analysis of tissue-specific translational patterns. (**a**): Heatmap clustering between tissues (cluster annotation can be found in Appendix A). The color code represents row Z-scores of normalized gene counts (log2CPM). The lettering on the left indicates cluster numbers. Top genes related to each cluster are listed on the right side of the heatmap. (**b**): tSNE clustering analysis indicating the different translational patterns observed in the brain compared to other tissues. tSNE was conducted using log2CPM normalized read counts. (**c**): Volcano plots of differentially translated gene (DTG) analysis for each tissue versus all other tissues. The number of genes in each comparison is presented in Appendix A. N = 3 biological replicates per group.

**Figure 5 ijms-26-00706-f005:**
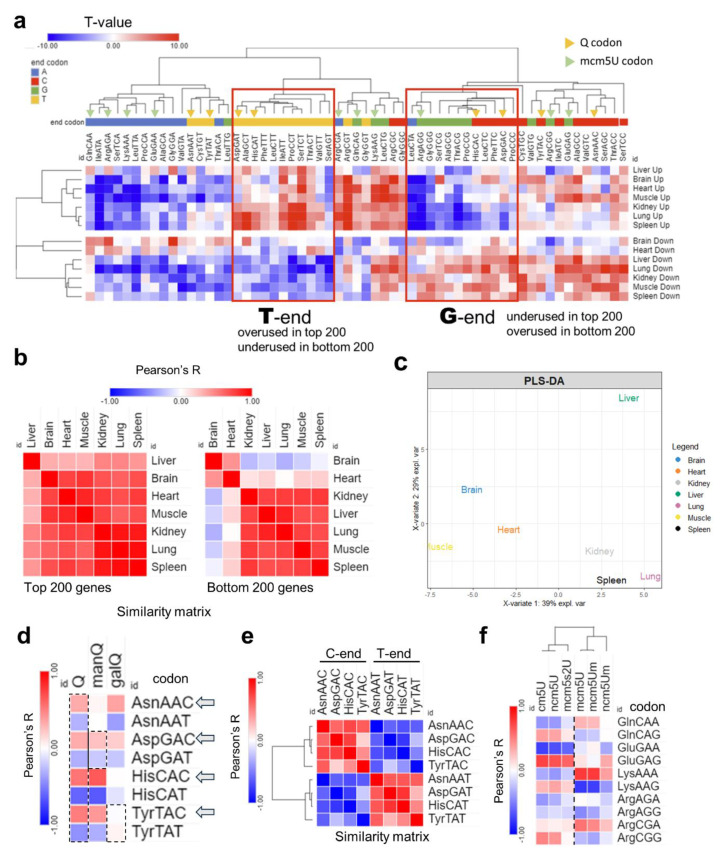
Analysis of isoacceptor codon usage. (**a**): Heatmap of isoacceptor codon frequency in each tissue. Isoacceptor frequency T-stats were calculated using top 200 (**up**) or bottom 200 (**down**) expressed genes from the Ribo-seq data in each tissue. Data presented as T-stat versus the genome average. T-stat of >2 or <−2 indicates *p <* 0.05. (**b**): Heatmap of Pearson’s correlation coefficients among 7 tissues by codon frequency from top (**left**) and bottom (**right**) 200 genes. (**c**): PLS-DA analysis of tissue clustering patterns using the top 200 genes. (**d**): Pearson’s correlation analysis between codon frequencies (using T-stat values of isoacceptors frequencies of each codon) and Q modification enrichment (Z-score of tRNA-Q from each tissue). (**e**): Pairwise correlation analysis of NAC vs. NAU Q codon from using T-stats as input. (**f**): Pearson’s correlation analysis of codon frequency, mcm5U, and related modification enrichment.

**Figure 6 ijms-26-00706-f006:**
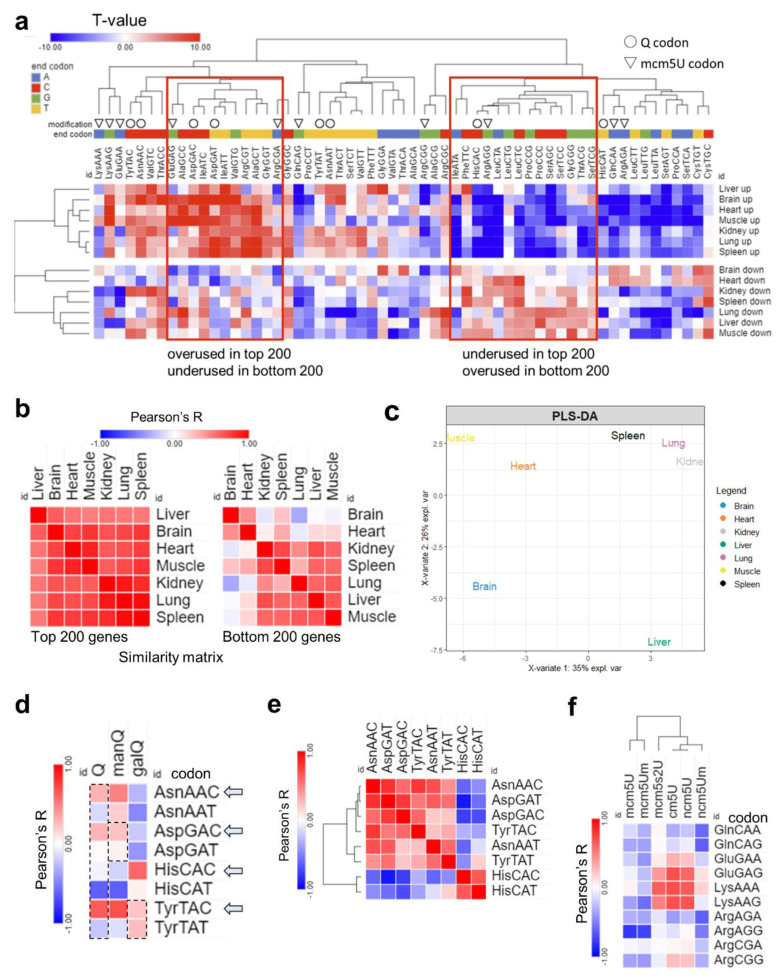
Analysis of total codon frequency. (**a**): Heatmap of total codon frequency in each tissue. T-stats were calculated using frequency of total codon counts using the same input as 4a. (**b**): Heatmap of Pearson’s correlation coefficients among 7 tissues by codon frequency from top (**left**) and bottom (**right**) 200 genes. (**c**): PLS-DA analysis of tissue clustering patterns using the top 200 genes. (**d**): Pearson’s correlation analysis between codon frequencies (T-stat of total codon frequencies) and Q modification enrichment (Z-score of tRNA-Q across tissues). (**e**): Pairwise correlation analysis of Q codon from Figure 5a. (**f**): Pearson’s correlation analysis of codon frequency, mcm5U, and related modification enrichment.

**Figure 7 ijms-26-00706-f007:**
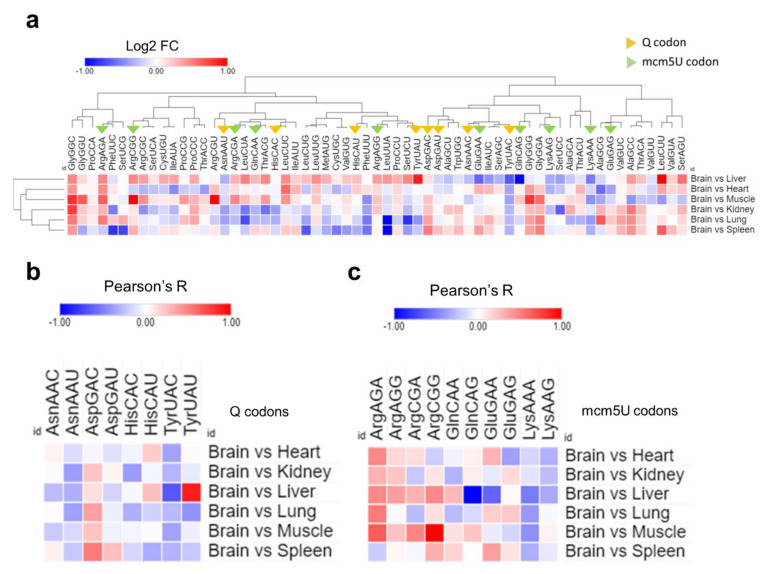
Analysis of A-site ribosome pausing. (**a**): Heatmap of A-site ribosome pausing analysis. The relative dwelling time in the brain compared to other tissues is represented as log 2 fold change (log2FC). (**b**,**c**): Selection of the codons related to Q modifications (**c**) or mcm5U modifications using (**d**) from Figure 8a for clarity.

**Figure 8 ijms-26-00706-f008:**
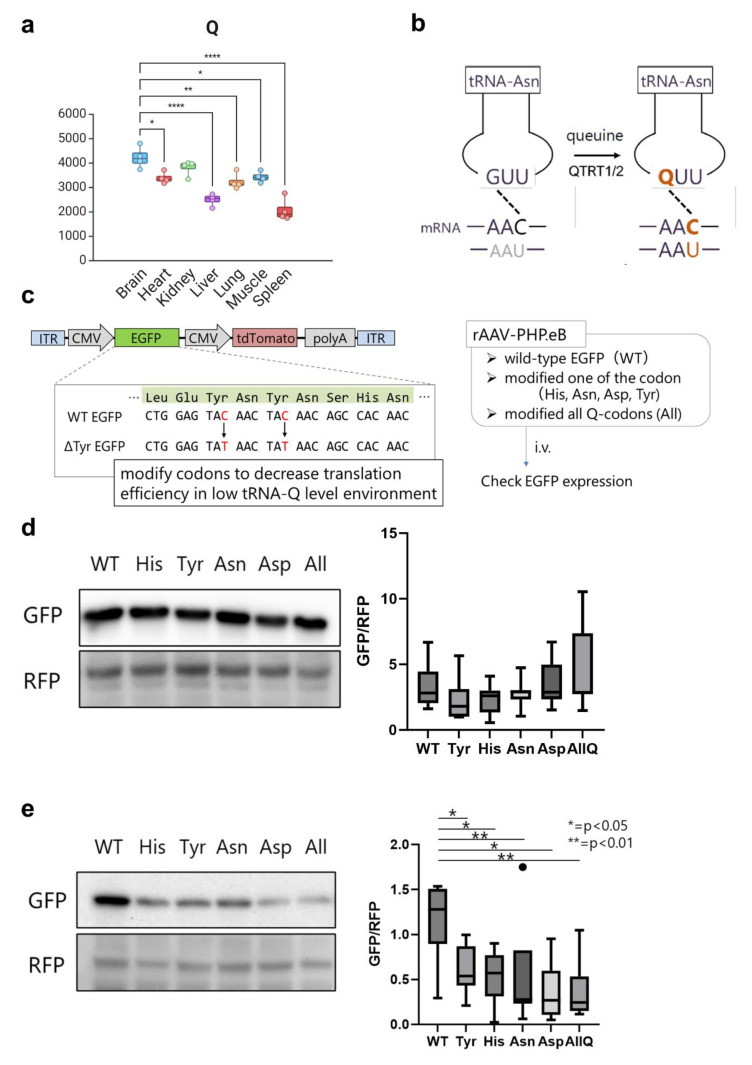
tRNA modifications inform codon optimization algorithms. (**a**): Expression of tRNA-Q across tissues. One-way ANOVA was conducted with Tukey’s post-hoc analysis. *: *p* < 0.05; **: *p* < 0.01; ****: *p* < 0.0001. Only comparisons between brain and other tissues are included in this graph. (**b**): Schematic showing how tRNA-Q expands the codon decoding of tRNA-Asn. (**c**): The design of the AAV vector carrying the codon-mutated EGFP. (**d**): Western blot of WT and mutant EGFP for each of the tRNA-Q decoded amino acids in the brain with quantification (densitometry). (**e**): Western blot of WT and mutant EGFP for each of the tRNA-Q decoded amino acids in the liver with quantification. Asterisks indicate statistical significance. N = 7 animals per group.

**Table 1 ijms-26-00706-t001:** Abbreviations, full names, and occurrences of examined tRNA modifications in this study. Chemical structures can be found elsewhere in the literature [4,35].

Abbreviation	Full Name	Occurrence in Small (<200nt) RNAs
**ac4C**	**N4-acetylcytidine**	**tRNA**
**acp3U**	**3-(3-amino-3-carboxypropyl)uridine**	**tRNA**
**Am**	**2′-O-methyladenosine**	**rRNA, snRNA, snoRNA**
**Cm**	**2′-O-methylcytidine**	**rRNA, snRNA, tRNA**
**cm5U**	**5-carboxymethyluridine**	**tRNA**
**D**	**5-carboxymethyluridine**	**rRNA, tRNA**
**f5C**	**5-formylcytidine**	**tRNA**
**f5Cm**	**5-formyl-2′-O-methylcytidine**	**tRNA**
**Gm**	**2′-O-methylguanosine**	**rRNA, snRNA, tRNA**
**hm5C**	**5-hydroxymethylcytidine**	**tRNA**
**hm5Cm**	**2′-O-methyl-5-hydroxymethylcytidine**	**tRNA**
**I**	**Inosine**	**tRNA**
**i6A**	**N6-isopentenyladenosine**	**tRNA**
**m1A**	**1-methyladenosine**	**rRNA, tRNA**
**m1G**	**1-methylguanosine**	**rRNA, tRNA**
**m1I**	**1-methylinosine**	**tRNA**
**m1Y**	**1-methylpseudouridine**	**unknown**
**m227G**	**N2,N2,7-trimethylguanosine**	**rRNA?**
**m22G**	**N2,N2-dimethylguanosine**	**tRNA, rRNA**
**m2G**	**N2-methylguanosine**	**rRNA, snRNA, tRNA**
**m3C**	**3-methylcytidine**	**tRNA**
**m3U**	**3-methyluridine**	**rRNA**
**m5C**	**5-methylcytidine**	**tRNA, rRNA**
**m5U**	**5-methyluridine**	**tRNA, rRNA**
**m5Um**	**5,2′-O-dimethyluridine**	**tRNA**
**m66A**	**N6,N6-dimethyladenosine**	**rRNA**
**m6A**	**N6-methyladenosine**	**rRNA, snRNA**
**m6Am**	**N6,2′-O-dimethyladenosine**	**snRNA**
**m6t6A**	**N6-methyl-N6-threonylcarbamoyladenosine**	**tRNA**
**m7G**	**7-methylguanosine**	**rRNA, tRNA, miRNA**
**man Q**	**Mannosyl-queuosine**	**tRNA**
**gal Q**	**Galactosyl-queuosine**	**tRNA**
**mchm5U_R**	**5-(carboxyhydroxymethyl)uridine methyl ester (R)**	**tRNA**
**mchm5U_S**	**5-(carboxyhydroxymethyl)uridine methyl ester (S)**	**tRNA**
**mcm5s2U**	**5-methoxycarbonylmethyl-2-thiouridine**	**tRNA**
**mcm5U**	**5-methoxycarbonylmethyluridine**	**tRNA**
**mcm5Um**	**5-methoxycarbonylmethyl-2′-O-methyluridine**	**tRNA**
**ms2i6A**	**2-methylthio-N6-isopentenyladenosine**	**tRNA**
**ms2t6A**	**2-methylthio-N6-threonylcarbamoyladenosine**	**tRNA**
**ncm5U**	**5-carbamoylmethyluridine**	**tRNA**
**ncm5Um**	**5-carbamoylmethyl-2′-O-methyluridine**	**tRNA**
**oHyW**	**Hydroxywybutosine**	**tRNA**
**Q**	**Queuosine**	**tRNA**
**t6A**	**N6-threonylcarbamoyladenosine**	**tRNA**
**Um**	**2′-O-methyluridine**	**rRNA, snRNA, snoRNA, tRNA**
**Y**	**Pseudouridine**	**rRNA, snRNA, snoRNA, tRNA**

## Data Availability

Raw sequencing data were deposited in the sequence read archive (small RNA-seq: PRJNA1003133. Ribo-seq: PRJNA1004093). All other data are presented in the Appendix A. Further data can be provided by the corresponding author upon reasonable request.

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
