# Peer review of "Decoding Codon Bias: The Role of tRNA Modifications in Tissue-Specific Translation"

_ijms, 2025, doi:10.3390/ijms26020706_

Round 1
Reviewer 1 Report
Comments and Suggestions for Authors
The authors revealed distinct enrichment patterns of tRNA modifications in tissues. For example, queuosine tRNA modification is most enriched in the brain compared to other tissues, while mitochondrial tRNA modifications and tRNA expression are highest in the heart. Furthermore, the authors have confirmed that the protein levels of mutant EGFP were downregulated in liver, which is poor in Q, while in brain EGFP levels did not change by using codons mutated EGFP for Q-codons. The results described in the manuscript are interesting and significant for many researchers to study on the tRNA epitranscriptome. However, I would like to give only a minor comment described below.
Minor comment
I would like to recommend to the authors to discuss including the reason, if possible, whether the tRNA epitranscriptome at the translational control level, leads mainly or complements simply protein synthesis, because I consider that the control of protein synthesis might be sufficient at the transcriptional level.
Author Response
The authors revealed distinct enrichment patterns of tRNA modifications in tissues. For example, queuosine tRNA modification is most enriched in the brain compared to other tissues, while mitochondrial tRNA modifications and tRNA expression are highest in the heart. Furthermore, the authors have confirmed that the protein levels of mutant EGFP were downregulated in liver, which is poor in Q, while in brain EGFP levels did not change by using codons mutated EGFP for Q-codons. The results described in the manuscript are interesting and significant for many researchers to study on the tRNA epitranscriptome. However, I would like to give only a minor comment described below.
Response: We would like to thank the reviewer for his very positive feedback.
Minor comment
I would like to recommend to the authors to discuss including the reason, if possible, whether the tRNA epitranscriptome at the translational control level, leads mainly or complements simply protein synthesis, because I consider that the control of protein synthesis might be sufficient at the transcriptional level.
Response: We do not think in the context of this work that tRNA epitranscriptome is the main driver of protein synthesis, rather, it would make sense that it is part of cellular adaptation to ensure proper physiological proteostasis. Nonetheless, in pathological conditions, most famously cancer, alterations in tRNA epitranscriptome are known to drive oncogenesis via codon biased translation. In lines 522-524 we state “In conclusion, we show that tRNA modifications expression levels in various tissues are fine-tuned to tissue specific codon decoding and mRNA translation requirements to maintain physiological proteostasis needed for tissue functioning
Reviewer 2 Report
Comments and Suggestions for Authors
The paper begins with 8 figures in a row without any discussion in the text. It would be better to discuss each figure separately as it is introduced in the text.
There are quite a few points when I feel unsure about the meaning. I think the paper needs some additional definitions and clarifications. For example, there is frequent talk of “enrichment of a modification”. But this could mean several things –
· The total quantity of tRNAs in a cell with a modification is increased.
· The fraction of tRNAs with the modification is increased relative to all other tRNAs
· The fraction of tRNAs with a the modification is increased relative to other tRNAs for the same amino acid with different anticodons.
· For one specific tRNA gene, the fraction of the molecules that have the modification is increased relative to unmodified molecules with the same anticodon.
These are all slightly different, and the paper should be precise about what is meant. Furthermore, there are statements like ‘a certain modification was enriched in the brain but not in some other tissue’. Here it is not clear whether the frequency in the brain is compared with the average of other tissues, or whether frequencies in each tissue are compared with some other null expectation based on gene frequencies, or equal expression rates of different genes.
In some cases there are genes with more than one different anticodon for the same amino acid, and these genes can be present in different numbers of copies. This kind of background information is needed before we can understand the current paper. For example, with the Q modification, I found data on the gene numbers from the following paper (Parisien M, Wang X, Pan T. Diversity of human tRNA genes from the 1000-genomes project. RNA Biol. 2013 Dec;10(12):1853-67. doi: 10.4161/rna.27361.)
Amino acid Anticodon # genes
Asn ATT 1
Asn GTT 23
Asp GTC 15
His GTG 10
Tyr ATA 1
Tyr GTA 12
The Q modification occurs at the first anticodon position of the GTN antocodons. In the cases of Asn and Tyr there are two different anticodons, so the relative frequencies of the Q modification could be changed by varying the expression rates of the two types of genes. It is mentioned (eg. line 224) that the expression rates are stable, but it is not clear how we know this. It also says (line 231) “We also analyzed the expression of tRNA-Q modified tRNAs. AspGTC was most enriched in the kidneys, while AsnGTT was equally enriched in both the Kidney and brain.” So are these variations due to variation of expression of genes between tissues or variation of rates of modifications of tRNAs? More clarifiction is necessary.
For all of the amino acids above, almost all the genes have G at the wobble position. So the more relevant factor is the fraction of molecules whose wobble G is modified to Q. I cannot find anywhere what this typical fraction is. There is a lot of discussion of enrichment of this fraction, but relative to what? What should this fraction be if there were no enrichment? Quantities presented as correlations and Z scores hide these fundamental numbers.
It is argued that there is some optimization of codon usage to match tRNA anticodons, and that this occurs differently in different tissues. The reasons for this are not clear. I can see that codon frequencies could differ between sets of genes that are highly expressed in different tissues. But what is the reason that the codon frequencies differ? Is there an evolutionary advantage to making them differ? Similarly, the tRNA frequencies (or modification frequencies) could differ between tissues, but what is the reason to make them differ? Are the tRNA frequencies adjusted to match the codons, or the codon frequencies adjusted to match the tRNAs? Are these two possibilities distinguishable?
The four amino acids above all have codons ending U or C. The only variable in the codon frequency data is the relative fraction of U to C codons. As there is only one variable, I would have thought that selection on codon usage should operate the same way in each case – i.e. highly expressed genes in each tissue should have the same codon frequencies, and highly expressed genes should have the same U to C ratio for each of the amino acids. Or at least, the direction of selection on codon usage (in favour of either C or U) should be the same, even if the strength of selection is different. If this is not the case, what is the reason? To what extent does the Q modification help with translation of both U and C codons? Is it really true that an unmodified G at the wobble position can only translate the C codon? Surely GU pairing is possible, even if the G is unmodified?
I think it is assumed here that tissues or genes differ in codon frequencies because of selection, but can it be ruled out that codon frequencies differ because of differences in mutational bias?
All the figures with Z scores and R values are hard to understand. There need to be clear formulae defining what data is used to measure the Z and R scores.
In summary, I am struggling to understand this paper. I have worked previously on evolution of codon usage and evolution of tRNA genes (although not in humans). I would expect to understand more than I do. My impression is that there is a lot of interesting data in this paper, but the meaning is just not coming through.
Author Response
We will begin by addressing the reviewer’s 2 comments from the final comment made.
In summary, I am struggling to understand this paper. I have worked previously on evolution of codon usage and evolution of tRNA genes (although not in humans). I would expect to understand more than I do. My impression is that there is a lot of interesting data in this paper, but the meaning is just not coming through.
Response: We appreciate the reviewer’s honestly in acknowledging his shortcomings in reviewing our work. We understand there are a lot of points that the reviewer might not be familiar with, and we will do our best to resolve them. We acknowledge that this work is somewhat challenging to interpret without knowing the relevant literature. And while the reviewer’s expertise in evolutionary tRNA biology are indeed relevant to understanding tRNA function, we believe our fields are different in how they approach tRNA and the tools that are used.
The paper begins with 8 figures in a row without any discussion in the text. It would be better to discuss each figure separately as it is introduced in the text.
Response: This was a typesetting issue that occurred after submitting the manuscript (we did not submit using the journal’s format). We corrected it in the revised manuscript.
There are quite a few points when I feel unsure about the meaning. I think the paper needs some additional definitions and clarifications. For example, there is frequent talk of “enrichment of a modification”. But this could mean several things –
- The total quantity of tRNAs in a cell with a modification is increased.
- The fraction of tRNAs with the modification is increased relative to all other tRNAs
- The fraction of tRNAs with a the modification is increased relative to other tRNAs for the same amino acid with different anticodons.
- For one specific tRNA gene, the fraction of the molecules that have the modification is increased relative to unmodified molecules with the same anticodon.
These are all slightly different, and the paper should be precise about what is meant. Furthermore, there are statements like ‘a certain modification was enriched in the brain but not in some other tissue’. Here it is not clear whether the frequency in the brain is compared with the average of other tissues, or whether frequencies in each tissue are compared with some other null expectation based on gene frequencies, or equal expression rates of different genes.
In some cases there are genes with more than one different anticodon for the same amino acid, and these genes can be present in different numbers of copies. This kind of background information is needed before we can understand the current paper. For example, with the Q modification, I found data on the gene numbers from the following paper (Parisien M, Wang X, Pan T. Diversity of human tRNA genes from the 1000-genomes project. RNA Biol. 2013 Dec;10(12):1853-67. doi: 10.4161/rna.27361.)
Amino acid Anticodon # genes
Asn ATT 1
Asn GTT 23
Asp GTC 15
His GTG 10
Tyr ATA 1
Tyr GTA 12
The Q modification occurs at the first anticodon position of the GTN antocodons. In the cases of Asn and Tyr there are two different anticodons, so the relative frequencies of the Q modification could be changed by varying the expression rates of the two types of genes. It is mentioned (eg. line 224) that the expression rates are stable, but it is not clear how we know this. It also says (line 231) “We also analyzed the expression of tRNA-Q modified tRNAs. AspGTC was most enriched in the kidneys, while AsnGTT was equally enriched in both the Kidney and brain.” So are these variations due to variation of expression of genes between tissues or variation of rates of modifications of tRNAs? More clarifiction is necessary.
Response: When we discuss enrichment, we mean relative levels of modifications. As we discuss in the various sections of the work, this could be partially explained by the levels of tRNAs in each tissue. For example, the spleen had lower tRNA content than other tissues and so on, so it showed lower enrichment of many modifications. Nonetheless, many modifications, as we show, cannot be explained by tRNA content. Rather, it appears to be a matter of stoichiometry (i.e. how many transcripts of tRNA carry the modifications from the total transcript pool). We have clarified what we mean by enrichment in the text at first mention to avoid confusion.
The number of tRNA genes have no role here. Especially, as we point out in the manuscript (line 452), that not all tRNA genes are transcribed into tRNAs. The genomic tRNA database is a good resource on this matter (Chan, P.P.; Lowe, T.M. GtRNAdb 2.0: an expanded database of transfer RNA genes identified in complete and draft genomes. Nucleic Acids Res 2016, 44, D184-189, doi:10.1093/nar/gkv1309). Thus, the number of tRNA genes is not a good measure to inform anticodon levels or tRNA expression. Here, we provide actual tRNA-seq results showing the expression levels of various tRNA anticodons. Thus, the argument of tRNA genes should not be considered, and the actual tRNA transcription is what should be a matter of focus. For example, While there is an Asn-ATT gene, that gene is not transcribed to a tRNA capable of contributing to translation.
Looking at tRNA transcripts, they could explain variations in modifications if the ratios between anticodons (isoacceptors) vary between tissues. However, in our work (Figure 2c) and in others works (Gao et al and Pinkard et al), isoacceptor expression is stable. Thus, variations in tRNA isoacceptor expression cannot explain differences in tRNA modifications levels, especially between tissues with comparable tRNA content such as the brain, heart, kidney, and the liver for example.
As for the statement in line 231 (related to figure 2d), we conducted this comparison by calculating the z scores across tissues. We acknowledged that this is a confusion approach and we removed figure 2d from the manuscript to make things easier to interpret.
We have added further explanation of these points in the manuscript (Lines 168~174)
For all of the amino acids above, almost all the genes have G at the wobble position. So the more relevant factor is the fraction of molecules whose wobble G is modified to Q. I cannot find anywhere what this typical fraction is. There is a lot of discussion of enrichment of this fraction, but relative to what? What should this fraction be if there were no enrichment? Quantities presented as correlations and Z scores hide these fundamental numbers.
Response: To assess the stoichiometry of Q modifications (i.e. the ratio of Q to G at position 34 in the 4 tRNAs carrying Q modifications) a different approach is needed. Either APM gel electrophoresis (which was shown by Cirzi et al, EMBO J, 2023, to range between 25~75% Q saturation) or periodate oxidation followed by sequencing (Katanski et al, Nucleic Acids Res, 2022). LC-MS/MS or regular tRNA-seq used herein cannot provide such information. We added this information to the discussion (Lines 486~488)
It is argued that there is some optimization of codon usage to match tRNA anticodons, and that this occurs differently in different tissues. The reasons for this are not clear. I can see that codon frequencies could differ between sets of genes that are highly expressed in different tissues. But what is the reason that the codon frequencies differ? Is there an evolutionary advantage to making them differ? Similarly, the tRNA frequencies (or modification frequencies) could differ between tissues, but what is the reason to make them differ? Are the tRNA frequencies adjusted to match the codons, or the codon frequencies adjusted to match the tRNAs? Are these two possibilities distinguishable?
Response: When one considers tissues such as the brain and spleen for example, the metabolic, bioenergetic, and functional needs are vastly different. Importantly, the molecular environment is not static, and the cells must adapt and respond to various environmental cues and adverse stimuli. Thus, we cannot expect neurons and solenocytes to have the same transcription and translational cues and machinery to respond to their environment. Why did it evolve to have different codons, that’s another question that we do not tackle here. I believe the expertise of the reviewer would be better suited to provide answers to this question as we are not evolutionary biologists. However, we believe that the tRNA modifications landscape, in the context of tissue enrichment, is an adaptation rather than a main reason for codon bias. That is, tissues need specific codons to translate specific genes, and so they need a specific set of modifications to help them with this task.
The four amino acids above all have codons ending U or C. The only variable in the codon frequency data is the relative fraction of U to C codons. As there is only one variable, I would have thought that selection on codon usage should operate the same way in each case – i.e. highly expressed genes in each tissue should have the same codon frequencies, and highly expressed genes should have the same U to C ratio for each of the amino acids. Or at least, the direction of selection on codon usage (in favour of either C or U) should be the same, even if the strength of selection is different. If this is not the case, what is the reason? To what extent does the Q modification help with translation of both U and C codons? Is it really true that an unmodified G at the wobble position can only translate the C codon? Surely GU pairing is possible, even if the G is unmodified?
Response: GU pairing does indeed occur, but it is inefficient. For example, in a preprint from our group (Rashad et al, 2024, doi: https://doi.org/10.1101/2024.02.14.580389) we show that after CRISPR/Cas9 mediated KO of Q modifications, there is significant ribosome stalling at U ending codons. In addition, Dixit et al (Nucleic Acids Res, 2021) showed that in the absence of Q, there is significant +1 frameshifting at the NAU codons, altering translation. Both these processes would render translation at NAU codons inefficient and would reduce the translation and protein levels of genes rich in NAU codons. We validate this hypothesis in figure 8 experimentally.
I think it is assumed here that tissues or genes differ in codon frequencies because of selection, but can it be ruled out that codon frequencies differ because of differences in mutational bias?
Response: We believe this is a question for evolutionary biologists that is outside of our expertise and the scope of this work. However, we hope our work will stimulate such thoughts and ideas in diverse fields.
All the figures with Z scores and R values are hard to understand. There need to be clear formulae defining what data is used to measure the Z and R scores.
Response: We added explanation to the z score and R in section 4.13 lines 687~696.
Reviewer 3 Report
Comments and Suggestions for Authors
The manuscript by Ando et al. provides an analysis of the composition of both isoaccepter and modification composition of tRNAs in different mouse tissues. They conclude that these parameters are optimized to support the differential gene expression in the various tissues. The experimental details are mostly well-described, and the experiments are carefully executed. The validity of the conclusion is as yet inconclusive because the authors were in a rush and forgot to include several figures. This final evaluation will have to await an updated version
Major issues:
- Add Figures 9-11.
- Define all abbreviations at first appearance.
- Please add a figure connecting the modification nomenclature to the chemical structures. Alternatively, reference a summary paper defining these parameters, e.g., doi: 10.1080/15476286.2020.1809197
- The authors intermingle “level of expression” with the “relative expression.” The data in the manuscript are all values of “relative expression” or relative composition. Without normalizing the results to sample size (amount of tissue entered into the processing) and the recovery of small RNAs during sample preparation, it is strictly not correct to refer to level, i.e., the absolute concentration. This is important because the concentration of each modified tRNA must be the parameter directly affecting the translation. As an example, the low relative abundance of tRNA in the spleen (Fig 2a) is either because the tRNA per cell mass is low or because the category “other” increases the total amount of small RNAs. The lack of distinction between these concepts makes the presentation confusing and imprecise.
- Lines 234-235: The abstract of reference 35 states, “We find that tRNA transcript levels vary substantially, whereas tRNA anticodon pools, which govern decoding rates, are more stable among cell types.” This conclusion is not accurately reflected in lines 234-235.
- Line 286: Since multiple other factors regulate gene expression, “drive” should be replaced with “contribute to.”
- Figure 4a: Define the lettering on the axes on either side of the diagram.
- Legend to Figure 4: Subfigure d is really subfigure d.
Author Response
The manuscript by Ando et al. provides an analysis of the composition of both isoaccepter and modification composition of tRNAs in different mouse tissues. They conclude that these parameters are optimized to support the differential gene expression in the various tissues. The experimental details are mostly well-described, and the experiments are carefully executed. The validity of the conclusion is as yet inconclusive because the authors were in a rush and forgot to include several figures. This final evaluation will have to await an updated version
Response: We thank the reviewer for his positive view of our work. We believe these was a misunderstanding of certain parts due to typesetting errors that led to the appearance that some figures are missing. We do apologize for this error.
Major issues:
- Add Figures 9-11.
Response: There are no figures 9-11 in the main text. These are supplemental figures that can be found in the online supplemental data. Supplementary figure 11 was incorrectly cited as figure 11 in the text. We apologize for this confusion.
- Define all abbreviations at first appearance.
Response: Abbreviations for tRNA modifications are explained in table 1. Other abbreviations were mentioned at first appearance.
- Please add a figure connecting the modification nomenclature to the chemical structures. Alternatively, reference a summary paper defining these parameters, e.g., doi: 10.1080/15476286.2020.1809197
Response: We provided citations in table 1, which lists the abbreviations and full scientific names of all tested modifications.
- The authors intermingle “level of expression” with the “relative expression.” The data in the manuscript are all values of “relative expression” or relative composition. Without normalizing the results to sample size (amount of tissue entered into the processing) and the recovery of small RNAs during sample preparation, it is strictly not correct to refer to level, i.e., the absolute concentration. This is important because the concentration of each modified tRNA must be the parameter directly affecting the translation. As an example, the low relative abundance of tRNA in the spleen (Fig 2a) is either because the tRNA per cell mass is low or because the category “other” increases the total amount of small RNAs. The lack of distinction between these concepts makes the presentation confusing and imprecise.
Response: We believe the reviewer is here referring to the LC-MS/MS data. As explained in the methods section (section 4.3), we use an LC-MS/MS with an in-line UV detector. The UV detector is set at 260nm wavelength. We use the signal from the UV detector to normalize the MS peaks, thus we can normalize the MS signal to the actual RNA injected into the machine. This eliminates errors occurring during various steps of preparation of the samples as well as LC-MS/MS injector errors. We did not mention absolute concentration in the manuscript as this requires a different approach with calibration curves that were done herein. Graphs for all modifications tested showing their normalized (corrected) peak areas are shown in S figures 2~8, including statistical analysis.
As for Fig2, while tRNA content is different between tissues (also validated by bioanalyzer analysis in supplementary fig 9), this has no impact on the ratio of tRNA isoacceptors (i.e. anticodons) shown in fig 2c.
Relating to the differences in tRNA content, we believe that the higher levels of snRNAs and snoRNAs in the spleen and lungs is what’s behind the lower relative tRNA content in these tissues. Nonetheless, as we explain in the text and as shown in the graphs is S figures 2~8, such differences cannot fully explain the tissue differences in tRNA modification levels.
Nonetheless, we amended our manuscript to clarify these issues.
- Lines 234-235: The abstract of reference 35 states, “We find that tRNA transcript levels vary substantially, whereas tRNA anticodon pools, which govern decoding rates, are more stable among cell types.” This conclusion is not accurately reflected in lines 234-235.
Response: If the reviewer goes into the details of the article by Gao et al, he will find that there is no disagreement. Gao et al refer by tRNA transcript levels to tRNA isodecoders. While tRNA anticodon pools (i.e. isoacceptors, shown in fig 2c) are stable. The same conclusion was made by Pinkard et al (Nat Comm, 2020). We did not examine isodecoders herein as it is not relevant to codon decoding. Thus, we made no comments on their expression. In summary, there is no disagreement once we examine the fine details of reference 35 (please see lines 432~435).
- Line 286: Since multiple other factors regulate gene expression, “drive” should be replaced with “contribute to.”
Response: We agree with the reviewer and made the edit.
- Figure 4a: Define the lettering on the axes on either side of the diagram.
Response: We added explanation in the legends.
- Legend to Figure 4: Subfigure d is really subfigure d
Response: This error was corrected.
Round 2
Reviewer 2 Report
Comments and Suggestions for Authors
The modifications have improved the paper. I now recommend acceptance.
Reviewer 3 Report
Comments and Suggestions for Authors
Thank you for addressing my comments. The paper is now a significant contribution